# Design of PI Fuzzy Logic Gain Scheduling Load Frequency Control in Two-Area Power Systems

**Tawfiq Hussein and Awad Shamekh \***

Electrical and Electronic Engineering Department, University of Benghazi, Benghazi-Libya,
P.O. Box 1308 Benghazi, Libya; tawfiq.elmenfy@uob.edu.ly
**\*** Correspondence: awad.shamekh@uob.edu.ly

**Abstract:** In this paper the use of the proportional integral (PI) algorithm incorporated with the fuzzy logic technique has been proposed as advanced gain scheduling load frequency control (GLFC) in two-area power systems. The proposed controller comprises two-level control systems, such that it consists of a pure integral compensator which is connected in parallel with a PI controller. However, and based on load demand, the PI parameters are updated online by means of fuzzy logic rules. With this control technique it becomes possible to eliminate steady state errors as well as to maintain good transient responses. The task of keeping a stable and overall satisfactory mode of operation in interconnected electric power systems is the main goal of any control strategy. This should be guaranteed over a wide range of operating conditions and particularly in sudden and drastic load changes. Therefore, the suggested approach has been examined following abnormal changes in loading conditions to clarify its reliability. The report also investigates the performance of the pure integral (I) controller and GLFC in individual configurations to highlight the advantages of the offered algorithm over the standard ones. The criterion of integral square error (ISE) has been exploited in the performance assessment for the designed controllers. Several simulation scenarios have been conducted, using the MATLAB–Simulink package, to illustrate the proficiency of the developed technique.

**Keywords:** two-area power system; load frequency control; conventional PI controller; fuzzy logic controller

## 1. Introduction

It has been reported intensively in the literature that frequency will be fixed in a power system if there is balance between the generated power and customer demand. The frequency of the power system mainly depends upon active power balance. Normally, there are many generators supplying power into the grid, these generators are supposed to be supervised with load frequency control (LFC) units to maintain the frequency at a preset value as well as to regulate the tie power line flow as planned.

Practically, if frequency drops at power plants, the automatic load-shedding should initiate the first stage to ensure the frequency is not lower than 49 Hz with a minimum of 10% to 20% of the rated load. The load frequency controllers at power plants should try to recover the frequency balance between power generation and power demand by increasing the megawatts of each generator to compensate the load demand. However, if the load frequency control failed to recover the frequency balance, and the frequency of generators drops to 47.5 Hz, the power stations are automatically tripped instantaneously, as the running of the power plants below this frequency becomes dangerous.

Typically, in many applications, conventional PI controllers form an essential component in the design of load frequency control. However, advanced tuning techniques are usually integrated in the

control law derivation to avoid the drawbacks of the standard PI controllers. Numerous advanced techniques have been introduced in the literature to accommodate this challenge. The work published in [1] summaries the most significant contributions in a comprehensive survey of the load frequency control techniques. The article classifies LFC into several categories. This includes: [2] Type of power system models; [3] control techniques; [4] control strategies; and [1] soft computing techniques

In the last decades, fuzzy logic control has become an attractive technique in power system applications and it has been implemented in various schemes by many researches. Chang and Fu [5] uses the fuzzy gain scheduling of proportional-integral (PI) controllers for a four-area interconnected power system with control dead bands and generation rate constraints. The fuzzy type controller was introduced in [6], where the upper and lower bounds of membership functions are obtained through a genetic algorithm. A genetic algorithm (GA)-based fuzzy gain scheduling was also proposed in [7], in which the PI controller gains are adaptively evaluated to reduce the burden of implementing a large number of fuzzy logic rules. The articles [8,9] proposed the fuzzy systems to tune the PI controllers. The papers considered the replacement of the conventional PI by the fuzzy PI controllers. The PI gains were essentially tuned online by the gain-scheduled fuzzy logic algorithm directly, without any need to identify the system model parameters periodically. This means a fast reaction according to the load change demand.

The work addressed in [10] presents modeling and simulating the interconnected two-area systems by means of the PI fuzzy controller with a sliding gain. The paper reports significant improvement in the performance specifications including the settling time and overshoot. Three control methodologies were developed in [11]. The article compares the performance of the conventional PI controller with an artificial neural network (ANN) and fuzzy logic controllers for three-area interconnected power systems. The study concludes the superiority of the advanced algorithms over the standard ones.

The neural network (NN) technique was also revised in [12]. The paper presents the idea of modifying the NN dynamics in different hidden layers of the power system's load frequency control. In the sensitivity considerations, the paper utilizes the NN emulator to identify the model and controller parameters simultaneously. A robust load frequency controller incorporated with a GA for a two-area interconnected power system was reported in [13]. The controller consists of two crisp inputs, namely the frequency deviation and its derivative. The output of the GA is then supplied as a control input to each area. The paper in [14] studied the control of load frequency in single- and two-area power systems with a fuzzy-like Proportional-Integral-Derivative PID controller. The study applied a multi-objective genetic algorithm to determine the controller parameters according to the system dynamics. The paper showed that the designed technique does its task much better than the traditional PID controller, which was tuned by Ziegler–Nichols method and the particle swarm optimization technique. Decentralized, the fuzzy logic load frequency controller that was suggested in [12] consists of two internal fuzzy logic controllers, including PD- and PI-like fuzzy controllers.

Recent articles have added significant contributions in this area of research, these include [15–17]. Gheisarnejad and Khooban [18] presents the fuzzy PD and cascade PI–PD controllers to accommodate the mismatch between the supply and load demand on micro-grids. The paper also introduces a modified JAYA algorithm to be utilized in the optimization solution problem. The sliding mode technique was proposed in [19] to sort out the issue of the time delay between the load demands and power generations in a stand-alone micro-grid system.

This article suggests using the fuzzy logic technique in a two-level control structure, which is designed in parallel with an integral control law. With this technique it is possible to address the issue of online controller adaptation as well as eliminating the frequency deviation. Following this introduction, the remainder of this report is organized to contain: the problem formulation of the proposed controller, the description of the two-area power system model, the simulation results, and the interpretations and conclusions.

## 2. Two-Area Power Systems

An incremental linearized model of the two-area power system, displayed in Figure 1, was undertaken in this study. This model structure is extensively used in the literature [8,9,20,21].

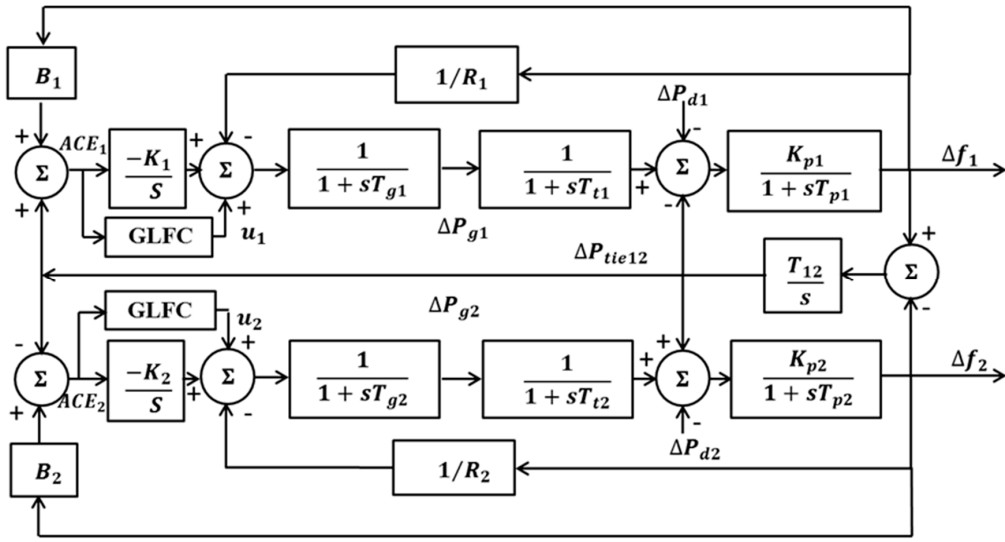

**Figure 1.** An incremental linearized model of two-area power systems.

The system parameters and loading conditions are listed in Table 1 [20]. This system can be expressed in a state equation as:

$$\dot{x}(t) = Ax(t) + Bu(t) + Ld(t) \tag{1}$$

and

$$u(t) = [u_1 \quad u_2]^T \tag{2}$$

$$d(t) = [\Delta P_{d1} \quad \Delta P_{d2}] \tag{3}$$

where *A*, *B*, and *L* are the state matrix, the input disturbance matrix, and the system disturbance matrix, respectively. $x(t)$, $u(t)$, and $d(t)$ are the state vector, the control signal vector, that is generated by GLFC, and the load change disturbance vector, respectively. The state vector is provided as

$$x(t) = [\Delta f_1 \ \Delta P_{g1} \ \Delta P_{d1} \ \Delta P_{tie12} \ \Delta f_2 \ \Delta P_{g2} \ \Delta P_{d2}]^T \tag{4}$$

where

*i* denotes the area number such that 1 is for area one and 2 is for area two,
$\Delta f_i$ is the frequency deviation,
$\Delta P_{gi}$ is the governor power deviation,
$\Delta P_{di}$ is the disturbance power deviation,
$\Delta P_{tiei}$ is the tie line power deviation.

The system output vector is denoted as

$$y(t) = \begin{bmatrix} \Delta f_1 \\ \Delta f_2 \\ \Delta P_{tie12} \end{bmatrix} \tag{5}$$

**Table 1.** The system parameters and loading conditions.

|  | Area 1 | Area 2 |
|---|---|---|
| Speed regulation | $R_1 = 0.05$ | $R_2 = 0.0625$ |
| Frequency sensitivity load coefficient | $D_1 = 0.6$ | $D_2 = 0.9$ |
| Inertia constant | $H_1 = 5$ | $H_2 = 4$ |
| Rated power | $P_1 = 1000$ MVA | $P_2 = 1000$ MVA |

The area control error is provided

$$ACE_i = \Delta P_{tie,i} + b_i \Delta f_i \tag{6}$$

where $b_i$ is the frequency bias constant. It is intuitive to consider a liner combination of the deviation in the tie line power and the frequency increment error to manipulate the system controller.

The system performance tracking index is characterized by *ISE* as:

$$ISE = \int e^2(t)dt \tag{7}$$

where $e$ is the output signals deviation.

## 3. Concepts of Gain Scheduling of the PI Controller Using Fuzzy Logic

The PID controller is a popular algorithm and is widely used in control problem solutions, specifically in industrial process control. It is called three terms, being Proportional-Integral-Derivative (PID) controller [22]. The text book algorithm has the following formula:

$$u(t) = K_p[e(t) + \frac{1}{T_i}\int_0^\tau e(t)dt + T_d\frac{d}{dt}e(t)] \tag{8}$$

where

$u(t)$ Control action
$K_p$ Proportional gain
$T_i$ Integration time
$T_d$ Derivative time

$$e(t) = y_{sp} - y_m \tag{9}$$

$y_{sp}$ Set point
$y_m$ Output measurement

The controller transfer function in the s-domain can be obtained by taking the Laplace transformation of Equation (8), this yields

$$U(s) = K_p E(s) + \frac{K_i}{s}E(s) + K_d s E(s) \tag{10}$$

or

$$U(s) = E(s)[K_p + \frac{K_i}{s} + K_d s] \tag{11}$$

However, in some circumstances, the derivative action requires specific tuning polices and advanced configurations in order to avoid severe consequences in several applications. Therefore, PI controller has become a popular structure particularly in design of LFC, where elimination of the frequency deviation represents the main target to be achieved.

### 3.1. Droop of the Generator (Speed Droop Governor)

The term "droop of the generator" in the power plant is the amount of frequency that is necessary to cause the power plant servomotor to change from fully closed to fully open [22]. In general, the droop of the generator (turbine mechanical speed) can be expressed in the following ratio:

$$R = \frac{\frac{-\Delta f}{f_n}}{\frac{\Delta P_g}{P_n}}\% \qquad (12)$$

$$\Delta f = f - f_n \qquad (13)$$

where $\Delta f$ is frequency deviation; $f_n$ is rated frequency; $P_g$ is active output power; and $P_n$ is rated active output power.

The governors have a higher speed ($N_o$) at no load than at 100% of the rated load ($N_r$). The regulation in percent of the governor is shown in Figure 2.

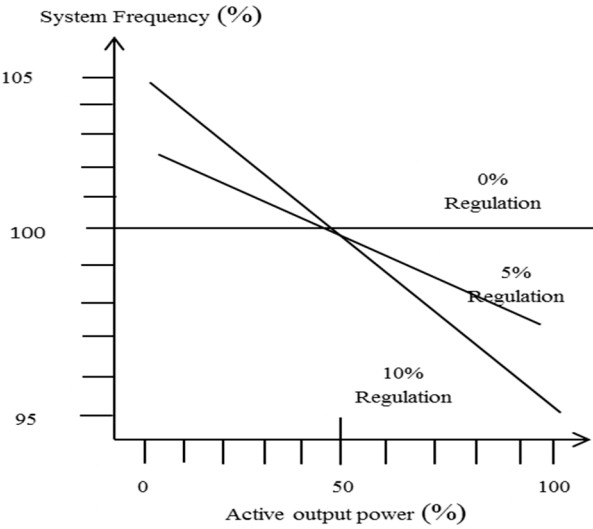

**Figure 2.** Regulation of the speed droop governors.

### 3.2. Speed Regulation ($R_p$)

The term of speed regulation refers to the amount of speed or frequency change that is necessary to cause the output of the synchronous generator to change from zero output to full output. In contrast, with droop of the generator, the speed regulation focuses on the output of the generator, rather than the position of power plant servomotors. In more details, if speed regulation ($R_p$), for example is 10% that means 10% speed change causes a MW change of 100% [14,23].

In this paper the proposed controller layout is the PI algorithm and it is well known that the success of the PI controller depends on an appropriate choice of the PI controller gains. The fuzzy logic control is incorporated as an advanced criterion of the PI controller tuning technique. The design determines the tuning rules base (IF-THEN Rules) for the PI gains by analyzing a typical response of the system, and then combine these rules into a fuzzy algorithm that is used to adjust the PI gains online. This method was introduced in [24] and further revised in this work.

As shown in Figure 3, the design considers two-level control structure, where the PI controller is the principle part and its gains are tuned online through Fuzzy system according to the lookup table given in Table 2 [25]. This part of the controller is named GLFC and connected in parallel with integral control law (I), which is tuned with fixed gain equal to 0.3 as proposed by [14]. With this strategy, two auxiliary feedback signals are added to the main feedback signal which is denoted by the speed regulation output. Therefore, the main feedback loop minimizes the increment of the frequency errors

quickly, and ultimately the supplementary feedback signals refine the system output responses and bring the errors to zero. Similarly, the philosophy of introducing GLFC with the fixed gain control action can be viewed as further improvement in the controller performance.

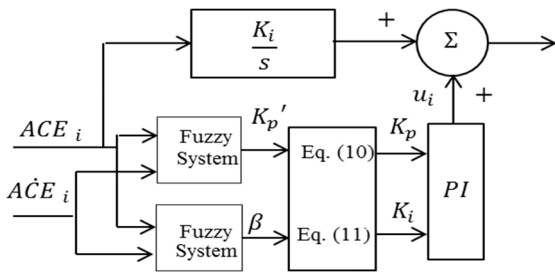

**Figure 3.** GLFC with two-level control structure.

**Table 2.** GLFC fuzzy rules.

| $\dot{ACE}_i$ / $ACE_i$ | BN | MN | SN | Z | SP | MP | BP |
|---|---|---|---|---|---|---|---|
| BN | BN | BN | BN | BN | MN | SN | Z |
| MN | BN | BN | MN | MN | SN | Z | SP |
| SN | BN | MN | MN | SN | Z | SP | MP |
| Z | MN | MN | SN | Z | SP | MP | MP |
| SP | MN | SN | Z | SP | MP | MP | BP |
| MP | SN | Z | SP | MP | MP | BP | BP |
| BP | Z | SP | MP | BP | BP | BP | BP |

Note that the meaning of symbols in the above table are as follows: B is big, M is medium, S is small, N is negative, P is positive, N is negative, and Z is zero.

The proportional gain, $K_p$, range is assumed to have the limits of $[K_{pmin}, K_{pmax}] \in R$. A new proportional gain, that complies with range constraints, is introduced as

$$K'_p = \frac{K_p - K_{pmin}}{K_{pmax} - K_{pmin}} \tag{14}$$

The integral constant can be also provided directly as a function of the proportional constant as

$$K_I = \frac{K_p}{\beta} \tag{15}$$

where

$K'_p$ and $\beta$ are the defuzzification outputs

The tuning criterion presumes that $K'_p$ and $\beta$, described in Equations (14) and (15), are the controller parameters to be updated by the fuzzy algorithm. The fuzzy IF-THEN rules are planned to match the following condition statement:

If $ACE\ \omega(t)$ is $A^i$ and $\dot{\omega}(t)$ is $B^i$ then $K'_p$ is $C^i$ and $\beta$ is $D^i$, where $A^i$, $B^i$, $C^i$, and $D^i$ are fuzzy sets, $i = 1, 2, \ldots, M$, and assuming that the domains of interest of $\omega(t)$ and $\dot{ACE}\dot{\omega}(t)$ are $[ACE_{M-}, ACE_{M+}]$ and $[\dot{ACE}_{M-}, \dot{ACE}_{M+}]$, respectively. As shown in Figure 4, seven symmetrical Gaussian memberships were generated with a standard deviation of 0.2 as fuzzy sets, whereas each input variable is denoted with seven labels, as described in Table 2, to cover all the possible states [20].

In this work several defuzzification methods were utilized. This includes centroid, bisector, middle of maximum algorithms, and others. However, and for this application, the bisector technique provides the best results. The bisector method is given by Equation (16):

$$v = f(\underline{x}) = \frac{\sum\limits_{l=1}^{M} \theta_i \prod\limits_{i=1}^{p} \mu_{F_i^l}(x_i)}{\sum\limits_{l=1}^{M} \prod\limits_{i=1}^{p} \mu_{F_i^l}(x_i)} \tag{16}$$

where $M$ is the number of rules and $p$ is the number of inputs. $\theta_l$ is the centroids of membership functions of the output corresponding to the $M$ rules. $\mu_{F_i^l}(x_i)$ is the membership function assigned to the $l$th linguistic variable in the $i$th rule.

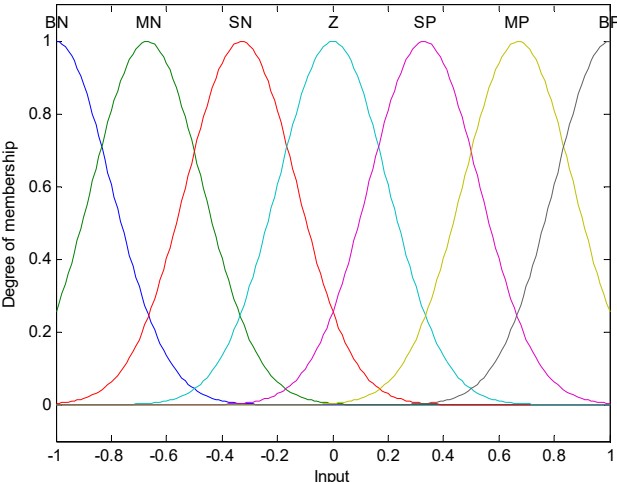

**Figure 4.** The seven memberships of symmetrical Gaussian functions.

## 4. Simulation Results

The MATLAB and Simulink package (R2013b) was exploited to perform the considered simulation scenarios. The simulation examples were carried out to compare the performance of GLFC technique with and without the integral action. Integral control law without GLFC was also included in the comparison. Figures 5–10 show the obtained results from the two-area system when it was subjected to load step changes of 187.5 and 250 MW, respectively, in area 1.

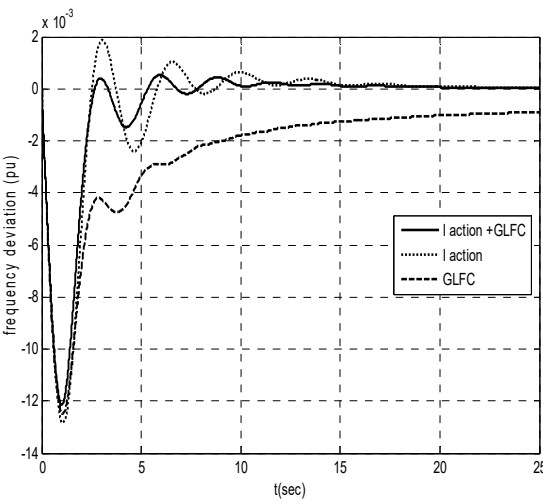

**Figure 5.** The frequency deviation response in area 1. (At step load change of 187.5 MW in area 1).

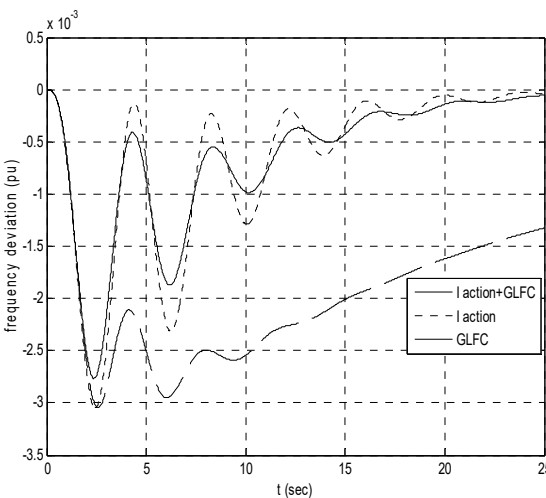

**Figure 6.** The frequency deviation response in area 2. (At step load change of 187.5 MW in area 1).

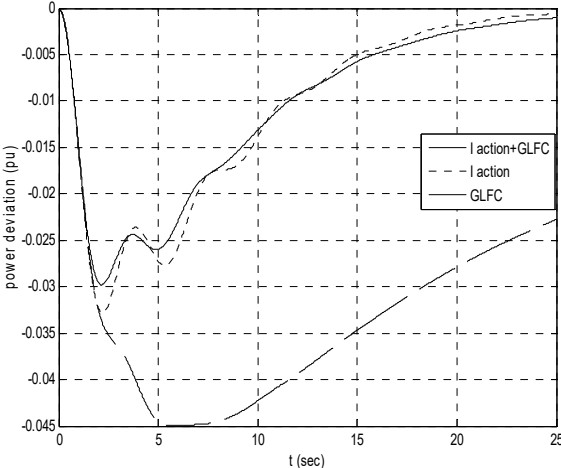

**Figure 7.** The tie line power deviation response. (At step load change of 187.5MW in area 1).

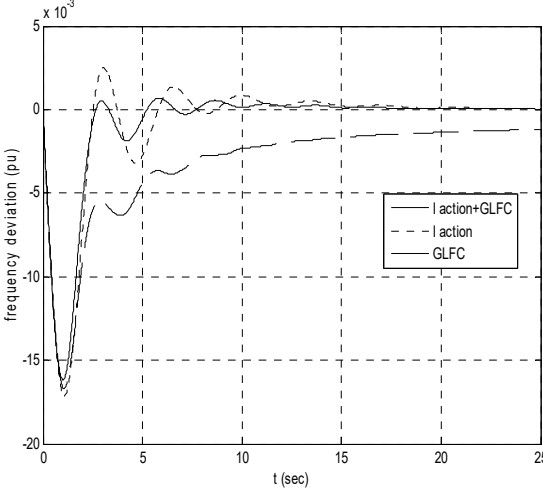

**Figure 8.** The frequency deviation response in area 1. (At step load change of 250 MW in area 1).

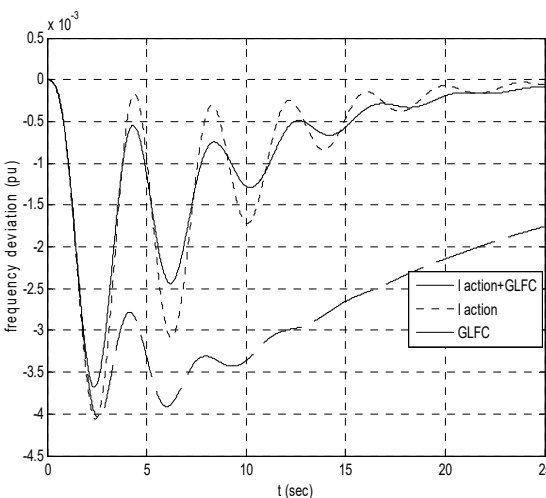

**Figure 9.** The frequency deviation response in area 2. (At step load change of 250 MW in area 1).

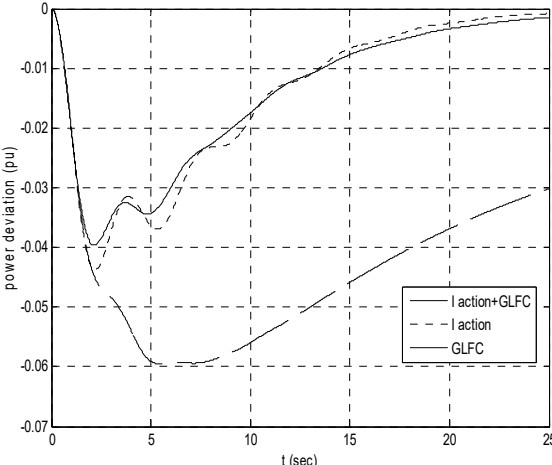

**Figure 10.** The tie line power deviation response. (At step load change of 250 MW in area 1).

From the simulation results and the performance index tables, it was obvious that the I+GLFC controller did the task better than the others. Several studies revealed that advanced fuzzy logic algorithms may be able to provide a satisfactory performance without aid of the standard ideal integral compensator. However, this usually requires further training with more complicated algorithms. Particularly, Figures 5–8 show that I+GLFC could eliminate the steady state error faster than I and GLFC, with less overshot.

With this control approach, the training burden was significantly attenuated and this actually comes from the benefit of employing GLFC in conjunction with the classic ideal integral component. In contrast, it was also noted that I-action is superior compared with GLFC when they work individually. This is a crucial inference and can be interpreted to the success of the integral control law compared with the fuzzy logic technique, which normally requires considerable effort to improve its performance.

The Tables 3–5 present the performance comparison of the three considered approaches in the sense of the ISE index. The results reveal that the I+GLFC technique provides a smaller ISE parameter compared with the other two techniques.

**Table 3.** The Performance index of $\Delta f_1$.

| Controller Type | ISE at Step Load Change of 187.5 MW in Area 1 | ISE at Step Load Change of 250 MW in Area 1 |
|---|---|---|
| I+GLFC | 0.0163 | 0.0290 |
| I | 0.020 | 0.0365 |
| GLFC | 0.0299 | 0.0582 |

**Table 4.** The Performance index of $\Delta f_2$.

| Controller Type | ISE at Step Load Change of 187.5 MW in Area 1 | ISE at Step Load Change of 250 MW in Area 1 |
|---|---|---|
| I+GLFC | 0.00212 | 0.00373 |
| I | 0.00257 | 0.00458 |
| GLFC | 0.0109 | 0.0194 |

**Table 5.** The Performance index of $\Delta P_{tie12}$.

| Controller Type | ISE at Step Load Change of 187.5 MW in Area 1 | ISE at Step Load Change of 250 MW in Area 1 |
|---|---|---|
| I+GLFC | 0.511 | 0.898 |
| I | 0.599 | 0.993 |
| GLFC | 3.65 | 5.36 |

## 5. Conclusions

Gain schedule PI fuzzy load frequency control (GLFC) in a two-level control approach was successfully applied for the load frequency control problem in two-area interconnected power systems. As it has been shown in this work, and likewise to [8,9], the performance of this technique does not need to identify the system model parameters periodically. In contrast, the controller parameters can be directly updated by means of gain scheduling, which can be inferred based on distribution of the selected fuzzy membership. With this approach it is possible to make the controller react quickly to follow up the load demand. In control theory it is well known that pure integral control is capable of eliminating the steady state error. However, it always results in a sluggish dynamic response. Therefore, GLFC is introduced with the integral control law to have a smooth response with zero steady state error.

The obtained results demonstrate that the proposed methodology provides the smallest ISE index in all simulation scenarios. Therefore, it is recommended to be considered in further and extended studies, and specifically in micro-grid systems.

**Author Contributions:** Conceptualization, T.H. and A.S.; methodology, T.H. and A.S.; software, T.H.; validation, T.H., A.S.; formal analysis, T.H., A.S.; investigation, T.H. and A.S.; resources, T.H. and A.S.; data curation, T.H. and A.S.; writing—original draft preparation, T.H. and A.S.; writing—review and editing, A.S. visualization, A.S.; supervision, T.H. and A.S.; project administration, T.H. and A.S.; funding acquisition, T.H. and A.S.

**Funding:** This research received no external funding.

**Conflicts of Interest:** The authors declare no conflict of interest.

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
