# Peer review of "Design of PI Fuzzy Logic Gain Scheduling Load Frequency Control in Two-Area Power Systems"

_designs, 2019_

Reviewer 1 Report

The variables in Fig. 1 and text aren’t consistent. I miss in Fig. 1 e.g. ΔPg , ΔPd

In Table. 2, first row: BN BN BN BN BM MN SN Z,     BM-is wrong

Fig. 6 - 10 different curves not possible distinguish, use solid, dash and dash-dot for curves.

I recommend add in references:

R. Venkata Rao,"Jaya:  A  simple  and  new  optimization  algorithm  for  solving  constrained  and

unconstrained optimization problems," International Journal of Industrial Engineering Computations, 2015

 T. BHARATH KUMAR & M. UMA VANI, "LOAD FREQUENCY CONTROL IN TWO AREA POWER SYSTEM USING ANFIS," International Journal of Electrical and Electronics Engineering Research (IJEEER)ISSN(P): 2250-155X; ISSN(E): 2278-943X, Vol. 4, Issue 1, Feb 2014, 85-92, © TJPRC Pvt. Ltd.

K.R.M.Vijaya Chandrakala and S. Balamurugan, "Adaptive Neuro-Fuzzy Scheduled Load Frequency Controller for Multi Source Multi Area System Interconnected Via Parallel AC-DC Links," International Journal on Electrical Engineering and Informatics - Volume 10, Number 3, September 2018        

Author Response

Many thanks for your valuable notes. All the comments have been considered(highlighted in yellow). Regarding the Figs we have already used  solid, dash and dash-dot. Would you Please try to maximize the figs (150%), and then it would very easy to see the different curves.

Reviewer 2 Report

 The authors used Proportional Integral  algorithm employing Fuzzy logic in two area power systems. Their aim is to reduce steady state errors and obtain good transient responses. Simulation results using Matlab-Simulink package show  the benefits of the developed technique. However, the paper would be strong if theoretical basis is given that guarantees the success of the algorithm in all conditions.

Author Response

Thank you for your comment. The paper aims to highlight that if an integral action is added to the PI-Fuzzy Logic control,  significant improvement can achieved. Therefore, I think it is clear that the advantage of the integral action (which is normally introduced to eliminate or minimize the steady state error) is combined with the standard PI-Fuzzy logic control to enhance the designed controller performance.  
